# Estimation of the Efficiency of Vessel Speed Reduction to Mitigate Gas Emission in Busan Port Using the AIS Database

**Donghan Woo and Namkyun Im *** 

Department of Navigation Science, Mokpo National Maritime University, Mokpo 58628, Korea; woodh@mmu.ac.kr
* Correspondence: namkyun.im@mmu.ac.kr; Tel.: +82-61-240-6311

**Abstract:** To mitigate marine atmospheric pollution causing greenhouse gas (GHG) and a threat to coastal residents' health in dense hub port cities, the Vessel Speed Reduction (VSR) programs were implemented in the Republic of Korea. Spatial analysis of the efficiency of the VSR programs is essential to monitor and improve the present programs. In the present study, the efficiency of the VSRs from the Busan Port (BP), including North Port (NP) and Gamcheon Port (GP), were analyzed. A bottom-up activity-based approach using automatic identification system (AIS) data was introduced herein for the estimation of spatial marine gas emission in real time. The BP has implemented the VSR program since 2020; thus, this study spatially analyzed marine gas emissions in the areas in 2019 and 2020 to demonstrate the efficiency of VSR. To demonstrate the VSR programs in the aspect of the comparison of gas emissions in each year, the total annual fuel consumption in each year is divided by the total arrival ships' GT respectively. According to the comparison of the spatial gas emission inventory between two years in the designed area, 19.2% of the annual marine gas emissions per ships' GT in BP in 2020 were reduced with the implementation of the VSR program. The spatial analysis clearly showed the mitigation of the ships' gas emissions along the passageway to BP. The spatial analysis of the efficiency of the VSR program will be useful basic data to reflect the local gas emission state on the improvement of the VSR program and newly established environmental regulations.

**Keywords:** air pollution; greenhouse gases (GHGs); ship gas emission; automatic identification system (AIS); spatial analysis; Vessel Speed Reduction (VSR)

## 1. Introduction

Of worldwide greenhouse gas (GHG) emissions, 2.89% was produced by shipping industries [1]. Accordingly, the International Maritime Organization (IMO) initialized a strategy to mitigate GHG produced from shipping industries in April 2018 [2]. The IMO [2] planned to reduce the total annual ships' GHG emissions by at least 50% by 2050 compared to GHG emissions in 2008. An Energy Efficiency Design Index (EEDI) was proposed by the IMO [3] to increase newly built ships' energy efficiency. To achieve the IMO's 2050 target, as mentioned before, actions for improving the operational efficiency of ships are also required [4]. Most of the port cities have been making an effort to increase energy efficiency by increasing prices and climate change mitigation with the reduction in GHG [5–7]. Air pollutants, such as CO, $NO_x$, $SO_x$, and PM, generated by shipping seriously impact residents' health problems in port cities [8]. Accordingly, there have been many studies regarding regulations to control air pollutants produced by shipping industries [9–11].

The IMO has been actively analyzing the gas emissions caused by shipping industries since 2018 to use this data for establishing a long-term plan to manage ship gas emissions in 2023 [12]. The European Union (EU) internationally leads environmental regulations controlling marine gas emissions. Regulations for ships over 5000 tons to annually report

and verify GHGs has been implemented since 2015 [12]. The EU plans to implement a system targeting trade systems to shipping industries if the IMO does not establish regulations to limit marine gas emissions until 2023 [12], emission control areas (ECAs), or sulfur emission control areas (SECAs) in which stricter controls were established to minimize ships' gas emissions specifically including $SO_x$, $NO_x$, ozone-depleting substances (ODSs), and volatile organic compounds (VOCs) since 2005 [13]. The Ministry of Ocean and Fisheries in the Republic of Korea has established Vessel Speed Reduction (VSR) programs to mitigate marine atmospheric pollution causing GHGs and a threat to coastal residents' health. These programs have been implemented in four major hub ports, such as Busan, Incheon, Ulsan, and Yeosu, since 2020. This program provides the benefit of a reduction in the cost of their port due to the voluntarily registered shipping companies, which annually demonstrates compliance with the VSR program with the required documents.

There have been many studies to precisely estimate marine gas emissions, such as the on-board survey of fuel consumption and action and field-testing and reviewing of ship emission factors [14–16]. In the past, the top-down approach was generally introduced to estimate the quantity of gas emission in a port using fuel sales for shipping [17–20]. This methodology has the limitation that it does not reflect the ships' movement in real time and, thus, the estimation of their gas emissions. Hence, the emissions estimation using the top-down approach has recently been considered to be uncertain [21]. To improve the reliability of the estimation of ship gas emissions, the bottom-up approach using automatic identification system(AIS) data was proposed. This approach could reflect each ships' different conditions in real time on the estimation's result [22]. Thus, the bottom-up approach could highly increase the accuracy of calculated shipping emission inventory [23–25].

Recently, analysis regarding ships' gas emissions in real time using the bottom-up approach has been demanded to enhance a sustainable climate and develop an air-quality management system [8]. Deniz et al. [26] introduced the bottom-up approach to estimate $NO_x$, $SO_2$, $CO_2$, hydrocarbon (HC), and particulate atter (PM) emissions from 7520 ships in the Gulf of Candarli in 2007 based on the consideration of the operation modes and types of ships. To determine the most affected regions from $NO_x$ emitted from ships, the spatial distribution of $NO_x$ was classified [27]. Ships' exhaust pollutants, such as $NO_x$, $SO_2$, and PM2.5, in the port of Piraeus in 2008 were estimated using AIS data [28]. The indicators to easily monitor the quantity of GHG emissions per ton of cargo or passenger was proposed by Villalba and Gemechu [29] using the theory of the bottom-up approach. Yau et al. [30] employed the bottom-up methodology using vessel specification in real time to develop a detailed maritime emission inventory for ocean-going vessels (OGVs) in Hong Kong with the base year of 2007. The marine emission costs of ships focusing mainly on a particular matter and volatile organic compounds were estimated using the bottom-up approach [31]. Based on AIS data, shipping gas emissions in Izmir Port in 2007 are calculated [32]. Annual emissions were estimated to analyze territorial ships' air pollution impact in the densely populated harbor [33]. The emission inventory caused by cruise shipping was estimated to analyze the social cost of cruise ship gas emission [34]. Tichavska and Tovar [35] used the bottom-up method to estimate gas emissions from cruise and ferry ships. The bottom-up approach was introduced to demonstrate the potential environmental influence of ship emission on the surrounding atmospheric environment along the coastline of the Yangtze River [36]. The ship activity-based methodology was introduced to calculate exhaust pollutant values (i.e., $NO_x$, $SO_2$, and PM2.5) during moving, maneuvering, and hoteling for international cruise ships from 18 ports of Greece in 2013 [37]. The spatial analysis of the gas emission inventory produced by liquid natural gas carriers in the world was carried out based on global AIS data [12]. Lee et al. [38] analyzed the atmospheric dispersion pattern of marine gas emission in the port of Busan in 2012 using the bottom-up approach and the CALPUFF model. Woo and Im spatially analyzed the marine gas emission in the port of Busan 2019 using the bottom-up approach with the territorial AIS data. An et al. [39]

verified the efficiency of VSR based on the panel data created by the official port control and air-quality measurement data in the big three port in the Republic of Korea.

The efficiency of the VSR program in the Busan Port (BP) was analyzed herein. In the present study, a bottom-up, activity-based approach using AIS data in real time is introduced to calculate spatial marine gas emission. To demonstrate the efficiency of VSR, this study spatially analyzed gas emissions from three types of ships, such as container, bulk carrier, and general cargo ships in the BP in 2019 and 2020, because the Ministry of Ocean and Fisheries in the Republic of Korea has implemented the VSR program since 2020. The analysis of the efficiency of the VSR programs will be useful basic data to reflect the local gas emission state on the improvement of the VSR programs and newly established environmental regulations.

## 2. Vessel Speed Reduction (VSR) Programs in BP

### 2.1. Detail of the VSR Programs in BP

The Ministry of Ocean and Fisheries in the Republic of Korea established the VSR programs for BP, including the North Port (NP) and Gamcheon Port (GP), as follows:

- The target ships of the VSR programs are the three types of ships, such as container ships, bulk carriers, and general cargo ships in arrival;
- The ships are over 3000 gross tonnage (GT);
- Voluntarily registered target ships for the benefit of the reduction in the cost of the port's dues need to reduce their speed to under 12 knots for container ships and car carriers and 10 knots for bulk carriers and general cargo ships;
- The VSR area is approximately 20 nautical miles from Yeong do Lighthouse and Oryukdo Lighthouse in the BP including the NP and GP. The detail of the pinpoints of the VSR area is shown in Figure 1.

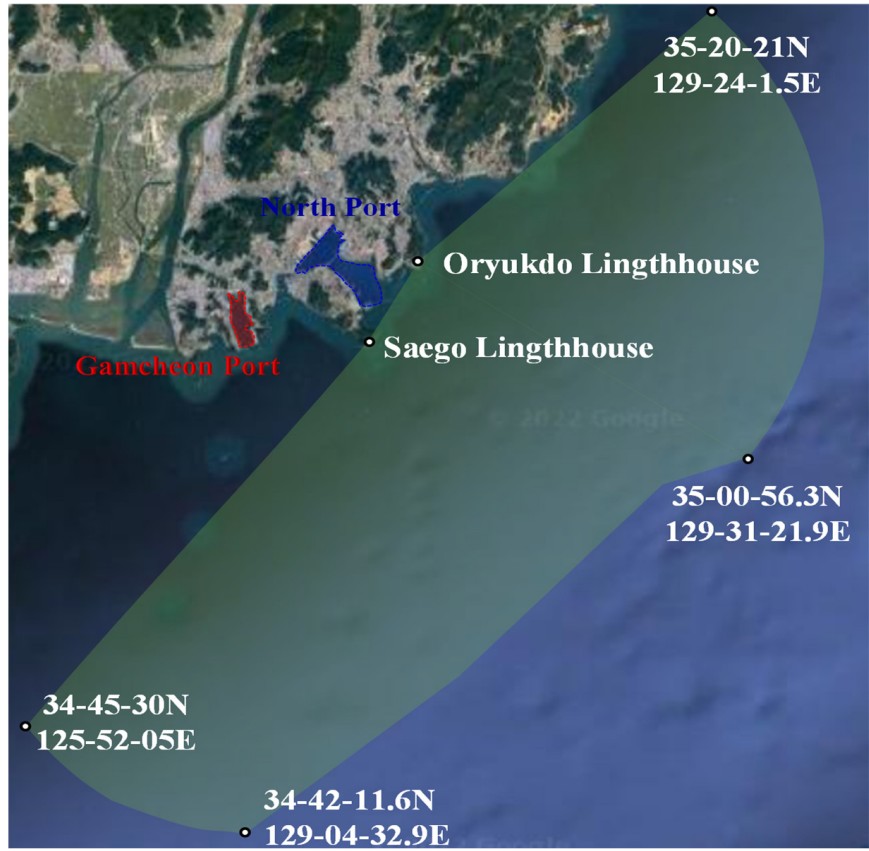

**Figure 1.** VSR program areas in BP including the NP and GP.

## 2.2. Rate of Compliance with VSR Programs in BP

Shipping companies that registered their vessels for the VSR program are required to submit documents according to the Port Management Information System (Port MIS) to prove whether they accurately comply with the program. The proposed calculation method from the Busan Port Authority (BPA) is as simple as dividing the ships' navigating distance by the total time in the designed areas of the VSR to calculate their average speed. Thus, the validation method needs to be improved to precisely monitor the ships' speed registered in the VSR area. The BPA reported the rate of compliance with the VSR programs in 2020, as shown in Table 1 [40], where it can be seen that most registered container ships successfully complied with the program at 96.3%. Relatively, the rate of compliance with the VSR program of bulk carriers was 66.6%, which is lower than other types of ships. Given that the total rate of compliance with the VSR program is 95.4%, in other words, most reported cases for the VSR program successfully complied with the required speed reduction in the designed area.

**Table 1.** Rate of compliance with the VSR programs in the NP and GP of BP in 2020.

| Type of Ship | Reported Cases | Improper Cases | Rate of Compliance with VSR Program |
|---|---|---|---|
| Container | 2769 | 103 | 96.3% |
| General cargo ship | 122 | 28 | 77.1% |
| Bulk carrier | 3 | 1 | 66.6% |
| Total | 2894 | 132 | 95.4% |

Source: Busan Port Authority [40].

## 3. Spatial Analysis Domain and AIS Data

### 3.1. Spatial Analysis Domain

To demonstrate the efficiency of the VSR programs, this study spatially analyzed marine gas emissions in the BP, including the NP and GP, in 2019 and 2020, because the Ministry of Ocean and Fisheries in the Republic of Korea has implemented the VSR programs since 2020. Figure 2 shows the analysis domain of the BP including the NP and GP [8]. Table 2 shows the details of the capacity of the BP. The BP has a total of 28 berths with a total berth length of 19,886 m. The NP services containers, general cargo ships, and bulk carriers with 18 berths that have a maximum capacity of 80,000 dead weight ton (DWT) class. Relatively, the GP has 10 berths for small and medium containers and general cargo ships. [8]. Approximately 6 million twenty-foot equivalent units (TEUs) of containers were handled in BP in 2019 and 2020 each year. The total number of BP's ship entry/departure in 2019 and 2020 was 93,701 and 89,018, respectively. BP is the second large city in the Republic of Korea and is completely surrounded by huge residential and industrial areas. Thus, a reduction in the arrival and departure of ships' gas emissions is necessary to protect the health of residents living in BP. Thus, the analysis of the efficiency of the VSR using spatial analysis according to AIS data will be a greatly useful supplement in the process of improving the VSR programs or the establishment of new environmental regulations.

**Table 2.** Details of the capacity of the BP [8].

| Port | Berths Length (m) | Port Service Capacity | | Type of Cargo |
| | | Ship DWT | Number of Berths | |
|---|---|---|---|---|
| North Port | 12,954 | 500–80,000 | 18 | Container, Passenger, General, Chemical, Oil, Raw (Sand, Fish) |
| Gamcheon Port | 6932 | 1000–50,000 | 10 | Container, General, Chemical, Oil, Fish |

Source: Busan Port Authority [40].

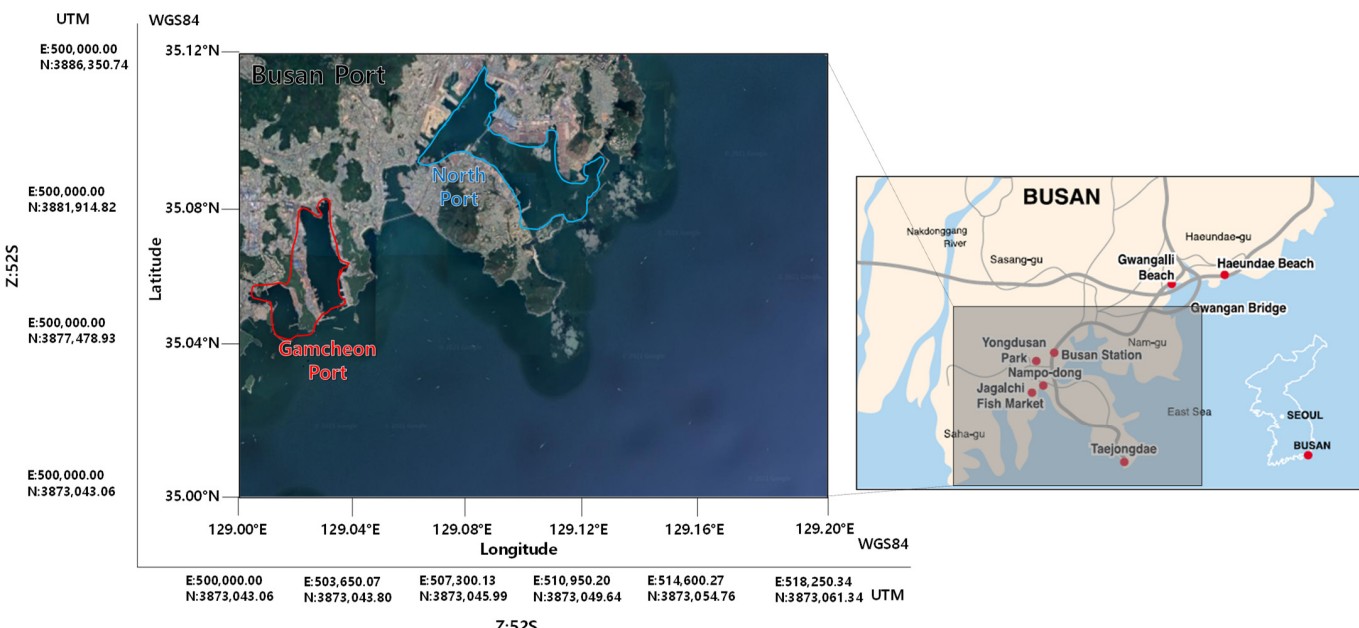

**Figure 2.** Analysis domain in BP [8].

### 3.2. AIS Data

IMO requires the installation of AIS on vessels that are over 300 GT engaged in international voyages and domestic cargo vessels of over 500 GT not engaged in international voyages to prevent collision accidents at sea. AIS transponders automatically locally broadcast ships' information in real time to share their operation status and movement with adjacent vessels and vessel traffic service (VTS). It communicates at intervals of 2–10 s while underway using the engine. Herein, AIS data in the status of underway using the engine is only used to demonstrate the efficiency of the VSR programs.

Preprocessing AIS Data

The AIS data were collected within the analysis domain of BP in 2019 and 2020 from MarineTraffic [41] herein. In the present study, the three types of ships (i.e., container ships, bulk carriers, and general cargo ships) and over 3000 GT were extracted from the original AIS data according to the target ships of the VSR program. Table 3 shows the total vessel arrivals for the three ship types in the NP and GP in 2019 and 2020 according to the AIS and the official records of the Korea Maritime Insititute (KMI). As per the official records of the KMI in Table 3, the total number of arrivals of the three types of ships over the two years were 26,393 and 25,536, respectively. Compared to the official records from KMI, the composition of the ship type based on the AIS data was similar. The differences in total vessel arrivals in two years between the two sources of data were 85 (+0.32%) and 83 (+0.21%), respectively. In other words, the territorial AIS data were successfully collected.

**Table 3.** Vessel arrival of containers, general cargo ships, and bulk carriers in BP in 2019 and 2020.

| Year | 2019 | | | 2020 | | |
| Type of Ship | Official Records | AIS Data | Difference | Official Records | AIS Data | Difference |
|---|---|---|---|---|---|---|
| Containers | 13,987 | 14,017 | 30 (+0.21%) | 12,974 | 13,001 | 27 (+0.21%) |
| General cargo ships | 11,358 | 11,404 | 46 (+0.41%) | 11,528 | 11,547 | 19 (+0.16%) |
| Bulk carriers | 1048 | 1057 | 9 (+0.58%) | 1034 | 1041 | 7 (+0.68%) |
| Total | 26,393 | 26,478 | 85 (+0.32%) | 25,536 | 25,589 | 53 (+0.21%) |

Source: MarineTraffic [41] and Korea Maritime Insitute [42].

Before using the AIS data, preprocessing of the AIS data was necessary, because the data were sometimes missed or wrongly transmitted with error values [8]. Some ships

transmitted their AIS information including an exceptionally high speed or were in a position where a ship physically could not be positioned [12]. Some ships' status indicated them as being underway using the engine, but they did not have a speed as the status but were moored or anchored instead, because their statues were manually changed by the ships' navigation officers. Sometimes they missed out to change the current statutes. Some ships' AIS data's draught, which is required to calculate the resistance of ships, was omitted. To use the bottom-up, activity-based approach during maneuvering, additional AIS information, such as ships' main engine force and age, is necessary for each ship's AIS item. AIS data for a ship while underway normally transmits a data every 2–10 s; however, some ships' AIS data were not properly transmitted to receivers. Thus, there were cases where the time between the two AIS data items of a ship were recorded as too long apparat, as over 10 s. This leads to a reduction in the reliability of the spatial analysis. Hence, this study conducted preprocessing of the AIS data as per the steps below:

- AIS items indicating their status as underway using their engine, but their speed as zero for over 1800 s were considered as not underway using their engine. They were deleted in the preprocessing;
- Of the total AIS data positioned on the landside or exceptionally deviated from the previous track, 2.8% were deleted;
- An exceptionally high speed was revised with an average of the previous and after the speed of a ship to reflect its characteristically intended speed on AIS data;
- Omitted ships' draught were inputted with their summer draught to conservatively reflect herein;
- AIS information, such as ships' main engine force and age, were added to each ship's AIS item;
- AIS data items were added by dividing the straight line of the linear interpolation between the two locations of AIS items in the interval of 10 s, and the average ship speed of the two AIS data items were inputted along the straight line. Figure 3 shows an example of the process.

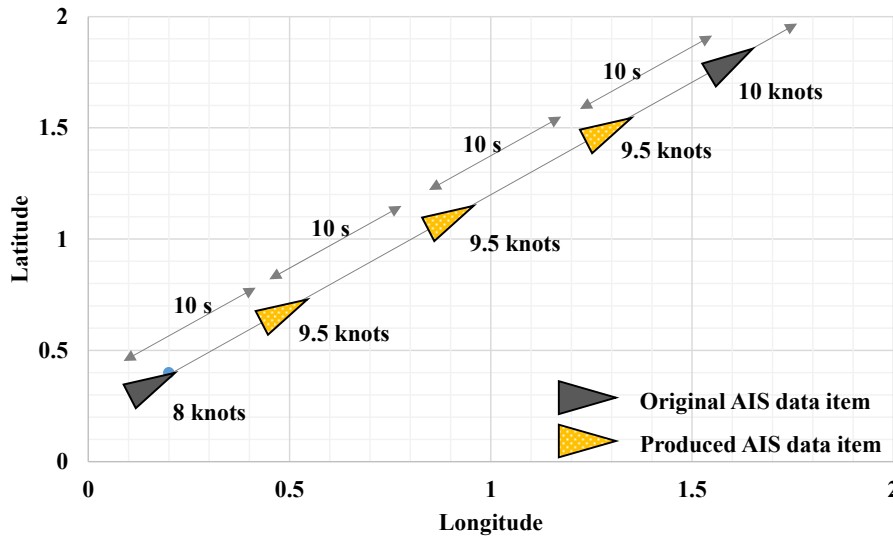

**Figure 3.** Example of the process of producing additional AIS data items.

## 4. Ship Gas Emission Calculation

### 4.1. Bottom-Up Approach to Calculating Ship Gas Emission

Figure 4 illustrates the process of the bottom-up methodology using AIS data to estimate gas emission of ships underway using their engine. Three items, such as ship's position, speed, and draught, in the AIS data were preprocessed as the first step to revise their error values [8]. The total resistance was calculated according to the proposed methodology by the International Towing Tank Conference (ITTC) as the second step [43,44]. The

fuel consumption was calculated according to the specific fuel oil consumption (SFOC). Table 4 shows SFOC by engine age. Table 4 shows that the efficiency of the engine decreases age [13]. The quantity of a ships' emissions is estimated based on the emission factors of each emission pollutant as shown in Table 5.

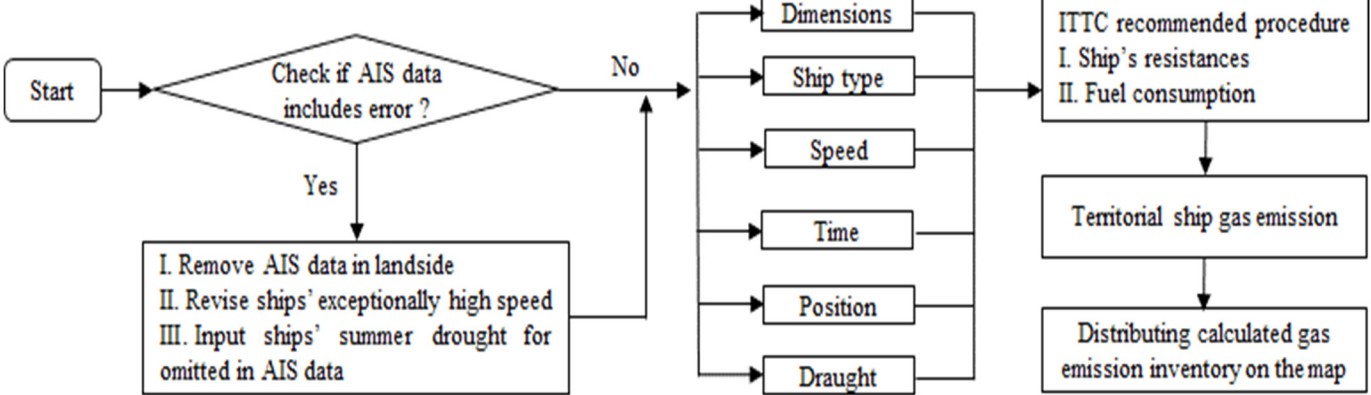

**Figure 4.** Procedure of the bottom-up approach using AIS data.

**Table 4.** SFOC (g/kW h) of main engines.

| Engine Age | Above 15,000 kW | 15,000–5000 kW | Below 5000 Kw |
|---|---|---|---|
| Before 1983 | 205 | 215 | 225 |
| 1984–2000 | 185 | 195 | 205 |
| 2001–2007 | 175 | 185 | 195 |

Souce: Second IMO greenhouse study 2009 [13].

**Table 5.** Emission factors of pollutants.

| Emission Pollutant | HFO Emission Factor (g/g fuel) |
|---|---|
| $CO_2$ | 3.11400 |
| CO | 0.00277 |
| $NO_x$ | 0.06512 |
| $SO_x$ | 0.04908 |
| PM | 0.00699 |

Source: Third IMO Greenhouse Gas Study 2014 [45].

### 4.2. Calculation of Gas Emission Underway Using Engine

Currently, the official standard for the estimation of gas emission from shipping industries does not exist in the Republic of Korea. Thus, Equation (1) suggested by US Environmental Protection Agency (EPA) [19,46,47] was introduced herein.

$$E_n = \sum_{i=1}^{n} VA_i \times P_{lj} \times LF_{ljm} \times T \times EF_{ljk} \tag{1}$$

where $E_n$ is gas emissions (kg) underway using a ship's engine; $VA_i$ is the total number of arrival ships; $P_{lj}$ indicates the average of the engine power (kW); $LF_{ljm}$ is the load factor (%); $T$ is the time (h); $EF_{ljk}$ indicates the emission factor (kg/kW h); $i$ is a single voyage; $j$ is the type of engine; $k$ is a type of pollutant; $l$ is the size of the ship (DWT); $m$ means the activity mode; $n$ is the total arrival ships.

Calculation of Installed Engine Power

Equation (2) was suggested to estimate the installed engine power to generate the desired speed [48].

$$P_{ij} = \frac{R_T \times V}{\eta_D \times \eta_T} + m \tag{2}$$

where $R_T$ means the ship's total resistance while navigating; $V$ is ship's speed (m/s); $\eta_D$ is the quasi-propulsion coefficient (typical range from 0.55 to 0.65) [49]; $\eta_T$ is the transmission efficiency (typical range from 0.95 to 0.98) [48]; $m$ indicates the sea margin which is typically 15–30% to reflect roughness, fouling, and weather on installed power depending on service route [48]. For the conservative estimation, $\eta_D$, $\eta_T$, and $m$ were assigned as maximum values in typical ranges of 0.65%, 0.98%. and 30%, respectively.

The total resistance of the ship, $R_T$, (KN) was calculated using Equation (3) proposed by ITTC herein [43].

$$R_T = 0.5 \times C_T \times \rho \times S \times V^2 \tag{3}$$

where $C_T$ is the total resistance coefficient; $\rho$ means the density of seawater; $S$ is the wetted surface.

## 5. Results and Discussion

To demonstrate the efficiency of the VSR in BP, including the NP and GP, this study spatially analyzed marine gas emission inventory (i.e., $CO_2$, CO, $NO_x$, $SO_x$, and PM) in the designed area in 2019 and 2020 based on the AIS database in real time. The spatial gas emission inventories in BP in two years are meaningful supplements to demonstrate the efficiency of the VSR and newly establish or improve proper local environmental regulations in the port.

### 5.1. Estimation of Fuel Consumption in BP in 2019 and 2020

Total fuel consumptions of three types of ships, such as container ships, general cargo ships, and bulk carriers, which are subjected to the VSR programs in the analysis domain in BP in 2019 and 2020, was calculated using formulas during the navigation statutes of maneuvering. To demonstrate the VSR programs in the aspect of the comparison of the fuel consumption in each year, the total annual fuel consumption in each year was divided by the total arrival ships' GT, respectively, as Equation (4):

$$FC_{GT} = \frac{FC_T}{GT_T} \tag{4}$$

where $FC_{GT}$ denotes the fuel consumption per ship's GT each year; $FC_T$ denotes the total annual fuel consumption in each year; $GT_T$ denotes the total arrival ships' GT.

As shown in Table 6, $FC_{GT}$ in 2019 and 2020 were 276,020.5 tons and 266,753.1 tons, respectively. In 2019, 9267.4 tons of fuel were consumed more than in 2020. However, $FC_{GT}$ in the two years was 0.000816 and 0.00645, respectively. In other words, the VSR programs led to a 20.9% reduction of $FC_{GT}$ of the three types of ships in the analysis domain in 2020 compared to the result in 2019.

**Table 6.** Fuel consumption of ships underway using the engine in BP in 2019 and 2020.

| Year | Fuel Consumption (ton/Year) | Total GT Annual Arrival Ships (ton/Year) | Fuel Consumption per Ships' GT | Area of Concentrated Fuel Concumption (ton/Year) |
|---|---|---|---|---|
| 2019 | 276,020.5 | 338,363,585 | 0.000816 | 58,599.2 (21.2%) |
| 2020 | 266,753.1 | 413,496,887 | 0.000645 | 50,603.1 (18.9%) |

In 2019, 21.2%, and, in 2020, 18.9% of the fuel consumption in BP were concentrated on the passageway to NP as shown in Table 6. Figure 5 clearly shows the concentrated fuel consumption at the passageway to the NP and GP in both years. This indicates that ships normally increased their speed along the passageway. In the view of the resolution along the passage of the NP and GP in Figure 5, the NP has a higher resolution than

NP. It clearly explains the higher traffic in the NP than GP according to the AIS data. As shown in Figure 5, compared to the results for 2019, the annual fuel consumption in most of the analysis domain areas in 2020 relatively decreased. It could be considered due to the efficiency of the VSR programs, because ships' speed was the main increase factor or the high fuel consumption. In the trend line of the monthly fuel consumption, there was no severe fluctuation in both years as shown in Figure 6. In particular, the monthly fuel consumption relatively decreased from October to December in both years.

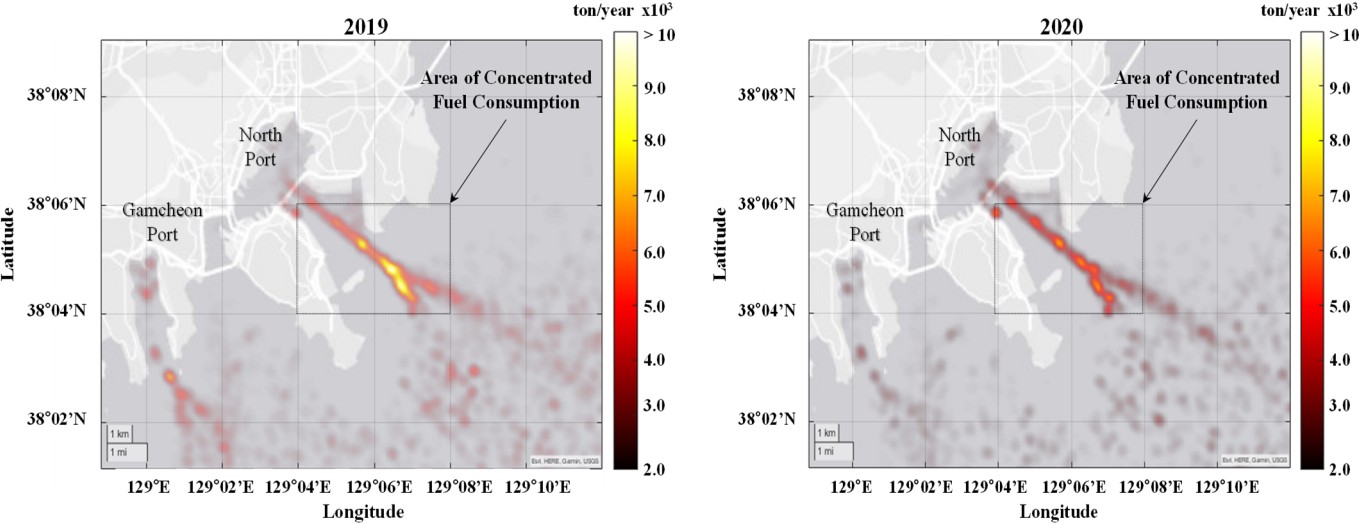

**Figure 5.** Spatial analysis of annual fuel consumption in BP in 2019 (**Left**) and 2020 (**Right**).

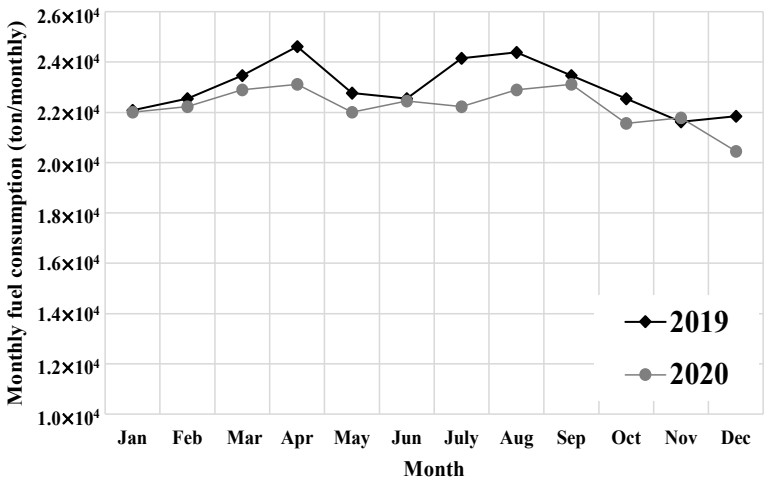

**Figure 6.** Monthly fuel consumption in BP in 2019 and 2020.

### 5.2. Estimation of Marine Gas Emission in BP

The estimated marine gas emission inventory in BP in 2019 and 2020 is presented in this section. Additionally, the estimated gas emission inventories were spatially illustrated on the analysis domain herein. These inventories included five types of air pollutants according to the estimated fuel consumptions in 2019 and 2020 as presented in the above section. Table 7 shows the amount of gas emission inventories of each air pollutant. Briefly, the total amount of the emitted from the three types of ships subjected to the VSR programs in BP in 2019 was 859,528 ton of $CO_2$, 765 ton of CO, 17,975 ton of $NO_x$, 13,547 ton of $SO_x$, and 1929 ton of PM as shown in Table 7 and Figure 7. The total amount emitted in 2020 was 830,669 ton of $CO_2$, 739 ton of CO, 17,371 ton of $NO_x$, 13,092 ton of $SO_x$, and 1865 ton of PM as shown in Table 7 and Figure 7.

**Table 7.** Marine gas emission inventory in BP in 2019 and 2020.

| Year | $CO_2$ | CO | $NO_x$ | $SO_x$ | PM | Total | Total GT of Annual Arrival Ships | Gas Emissions Per Ships' GT |
|---|---|---|---|---|---|---|---|---|
| 2019 | 859,527 | 765 | 17,975 | 13,547 | 1929 | 893,743 | 338,363,585 | 0.0026 |
| 2020 | 830,669 | 738 | 17,372 | 13,093 | 1864 | 863,736 | 413,496,887 | 0.0021 |

(unit: ton/year).

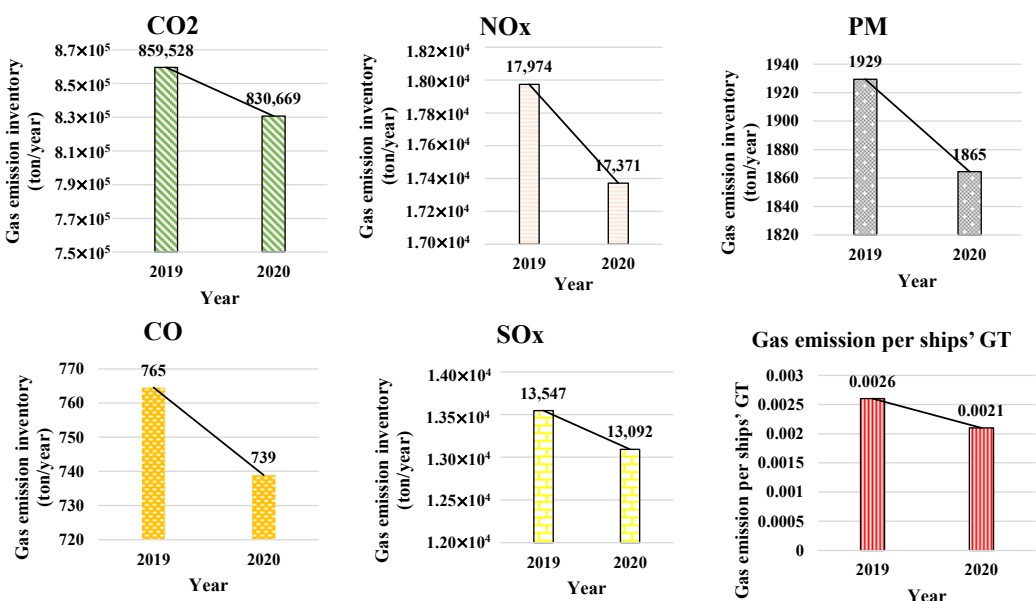

**Figure 7.** Gas emission inventory and gas emission per ships' GT in 2019 and 2020.

Figure 7 clearly shows the reduction in the amount of ship gas emissions in 2020 compared to 2019. According to the AIS data in BP, the difference in the number of the total arrival ships between 2019 and 2020 was 889; approximately 3.3% was reduced compared to the previous year as shown in Table 3. In terms of the rate of the reduction in the total gas emissions between the two years, it was approximately 3.4%. The two reduction rates of 3.3% and 3.4% were similar. In other words, the reduction in the number of the total arrival ships in BP led to the decrease in the total gas emissions in 2020 compared to the result of 2019. However, in the view of the forward, it is hard to demonstrate the efficiency of the VSR programs in BP.

To demonstrate the VSR programs in the aspect of the comparison of the gas emissions in each year, the total annual gas emissions in each year were divided by the total arrival ships' GT, respectively, as per Equation (5):

$$GE_{GT} = \frac{GE_T}{GT_T} \tag{5}$$

where $GE_{GT}$ denotes gas emission per ship's GT each year; $GE_T$ denotes the total annual gas emission in each year; $GT_T$ denotes the total arrival ships' GT.

Compared to $GE_{GT}$ in 2019, the efficiency of the VSR programs was 19.2% to reduce the $GE_{GT}$ in 2020. In other words, the VSR programs successfully mitigated arrival and departure ships' gas emissions in BP. Thus, it is expected that the extension of the VSR programs will considerably mitigate the total gas emission caused by ships in BP in the future.

Figure 8 illustrates the annual spatial distribution of five air pollutants in BP in 2019 and 2020. The gas emissions of the five air pollutants are roughly distributed along the berthing line and the passageway of BP in both years. Comparing two ports between the NP and the GP, it can be easily confirmed that the gas emissions were more distributed at the NP, because the main role of the GP is for fishing vessels, reefers, and relatively small

general cargo ships. In terms of the high concentration areas of the gas emissions in both years, the five air pollutants were highly distributed along the passageway between the port to the ocean at the NP, because ships increase their speed along the passageway during arriving or denaturing in the BP. In terms of these results, an improvement of the VSR programs is required to mitigate the concentrated gas emission along the passageway.

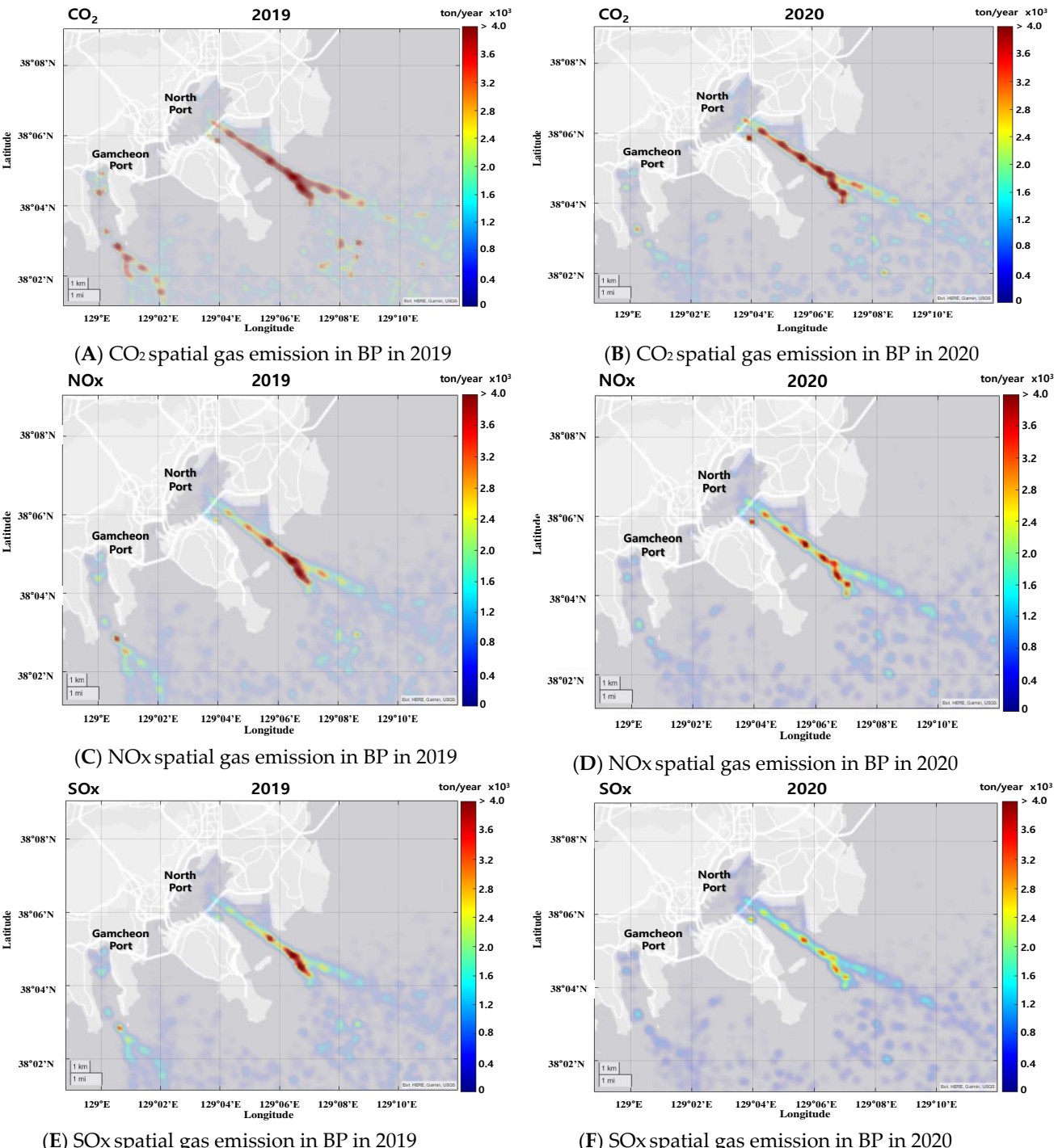

(**A**) CO2 spatial gas emission in BP in 2019

(**B**) CO2 spatial gas emission in BP in 2020

(**C**) NOx spatial gas emission in BP in 2019

(**D**) NOx spatial gas emission in BP in 2020

(**E**) SOx spatial gas emission in BP in 2019

(**F**) SOx spatial gas emission in BP in 2020

**Figure 8.** *Cont.*

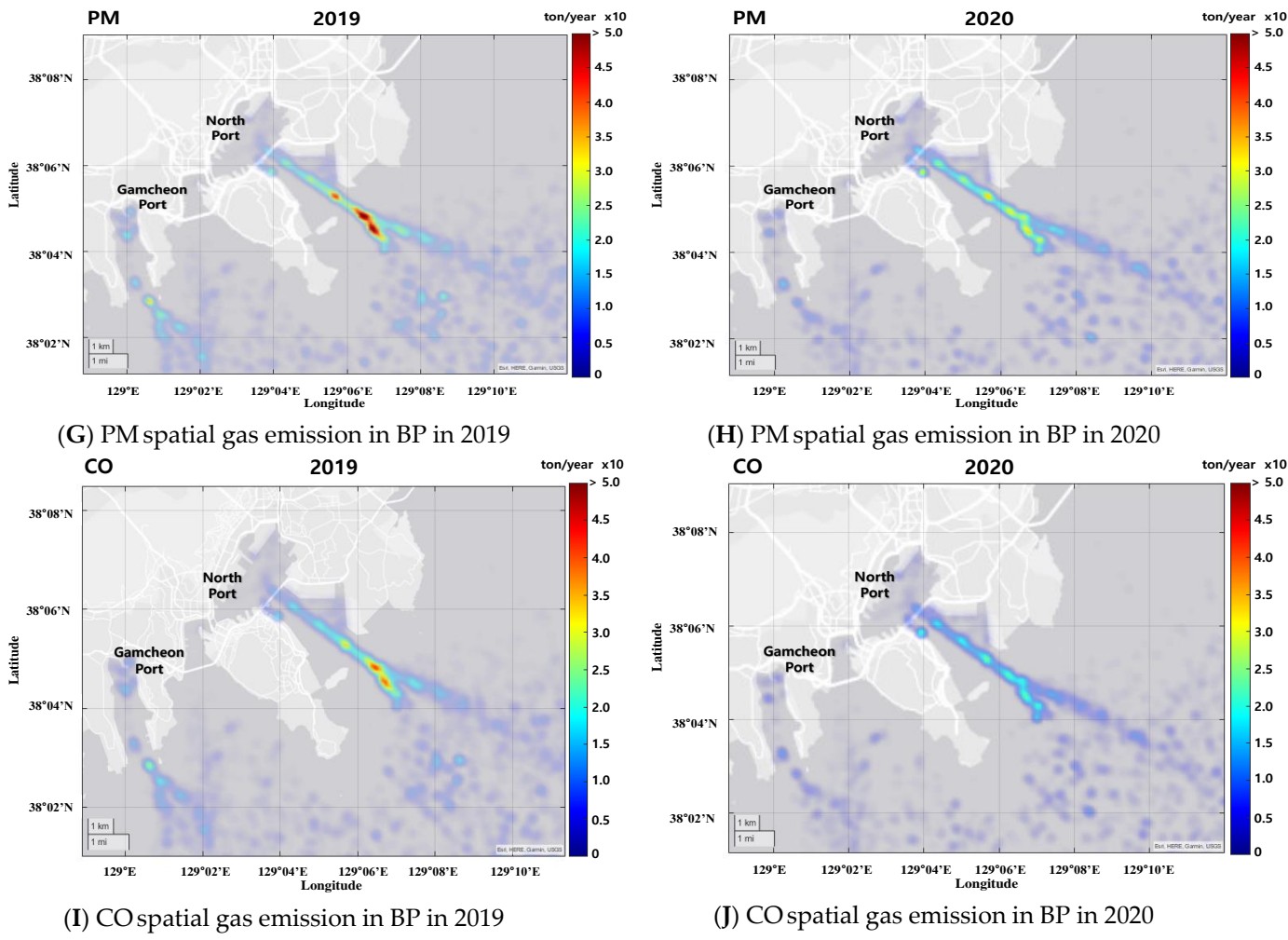

**Figure 8.** Comparison of the spatial analysis of the five air pollutants in BP in 2019 and 2020.

Compared to the previous year, 3.4% and 19.2% reductions of the amount of the total gas emissions and $GT_T$ in 2020, respectively, are spatially illustrated in Figure 8. The VSR programs successfully mitigated the spatial gas emissions in the analysis domain of BP as shown in Figure 8. Based on this spatial analysis, the atmospheric monitoring systems should be installed at the most optimal locations to track the achievement of the VSR program and exactly analyze its impact on the port.

## 6. Conclusions

This study demonstrated the efficiency of the Vessel Speed Reduction (VSR) programs to mitigate the ship gas emissions in BP in the Republic of Korea according to the comparison between the estimated gas emission in 2019 and 2020. The bottom-up, activity-based approach using the territorial automatic identification system (AIS) database was introduced to calculate the spatial gas emissions of five air pollutants in real time over two years.

Basically, compared to the results of 2019, the total amount of the five air pollutants generated by three types of ships in BP was reduced by 3.4% according to the reduction of the total arrival ships by 3.3%. In particular, this study demonstrated the efficiency of the VSR as 19.2% as the gas emissions per ships' GT according to calculated gas emissions inventories in 2019 and 2020. The spatial analysis on the map clearly shows the efficiency of the program according to the comparison of the density of the gas emissions in the analysis domain. Plotting ship gas emissions inventories on the map of the analysis domain gives a deeper understanding of the characteristics of the ship fuel consumption and gas emissions

as most of them are concentrated along the passageway to NP. It is because ships highly increase their speed along the passageway during arriving or departing port. Thus, the establishment of regulations for speed limits for the inner port of BP should be extensively designated. The gas emission in the inner port could lead to serious health problems, such as respiratory and cardiovascular diseases, reproductive and central nervous system dysfunctions, and cancer, for residents living in the vicinity.

Based on this spatial analysis, atmospheric monitoring systems should be installed at the most optimal locations to track the achievement of the VSR programs and exactly analyze their impact on the port. An improvement in the skills of the preprocessing of AIS data is necessary to increase the reliability of the bottom-up approach in future studies. To validate the estimation of ship gas emissions in the port, an introduction of a monitoring system of ships' gas emissions in real time in a specifically designed area in the port is required in the future. Additionally, the atmospheric diffusion pattern of the gas emissions considering weather data is required to categorize territories that are more highly to be affected by marine air pollutants.

**Author Contributions:** Conceptualization, software, data curation, investigation, writing—original draft, and writing—review and editing: D.W.; conceptualization, investigation, methodology, validation, and supervision: N.I. All authors have read and agreed to the published version of the manuscript.

**Funding:** This research received no external funding.

**Conflicts of Interest:** The authors declare no conflict of interest.

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
