# Peer review of "Estimation of the Efficiency of Vessel Speed Reduction to Mitigate Gas Emission in Busan Port Using the AIS Database"

_jmse, doi:10.3390/jmse10030435_

Round 1

Reviewer 1 Report

Comments to the Authors 

Journal: Journal of Marine Science and Engineering.
Manuscript ID: jmse-1617785

Title: Estimation of the Efficiency of Vessel Speed Reduction to Mitigate Gas Emission in Busan Port using AIS Database

Authors: Donghan Woo and Namkyun Im 2*

This manuscript proposed the Spatial analysis of the efficiency of VSR programs is essential to monitor and improve the present program. In the present study, the efficiency of VSR from the Busan Port (BP) including North Port (NP) and Gamcheon Port (GP) was analyzed. A bottom-up activity-based approach using Automatic Identification System (AIS) data is introduced herein for the estimation of spatial marine gas emission in real-time. These authors have proposed the spatial Analysis Domain, AIS Data analysis, and Ship Gas Emission Calculation by Bottom-Up Approach to determine the efficiency of Vessel Speed Reduction, which seems very popular and makes sense.

All of the Figures have to be fully written and modified for readers and reviewers easy to understand and review.

  1. Figures 1,2,3,5 and 8 should be modified with higher resolution and redrawing these figures are necessary needed.
  2. More descriptions of analyzing and preprocessing of AIS Data are needed to avoid similar to a short report and manuscript.
  3. The methodology of the bottom-up activity-based approach needs to be described in the paper review and add the evidence of other earlier papers analyzed.
  4. The topic of this manuscript is very popular, therefore, the core contribution of this manuscript has to propose more information to enhance the core value of this study.
  5. All of this construction of this manuscript is to be revised for detailed information and more explanation for readers easy to read.

Author Response

Thank you for your valuable comments to improve the paper. 

According to the reviewer's comments, the authors review the paper. 

Reviewer 2 Report

The paper deals with a very important and timely topic. My comments and suggestions are given below.

  1. Lines 43-45 - The sentence is repeating; please remove it.
  2. Lines 51-52 - Acronyms ODS and VOC used but not written in full.
  3. Line 55 - It is written "Busa"; should it be Busan?
  4. Line 73 - Acronyms HC and PM used but not written in full.
  5. Line 99 - Acronym BP is used, but not written in full. Although it is written in the Abstract, I suggest writing it in full in the text as well.
  6. I suggest referencing a recent paper that dealt with the implementation of the VSR programme: "Effects of a Vessel Speed Reduction Program on Air Quality in Port Areas: Focusing on the Big Three Ports In South Korea"; https://doi.org/10.3390/jmse9040407.
  7. Lines 102-103 - the study included three ship types, namely container ships, bulk carriers, and general cargo ships; however VSR program mentioned car carriers as well.
  8. Lines 104-108 - I suggest placing this part in the Discussion or Conclusion section.
  9. Line 114 - Acronyms NP and GP are used but not written in full (same comment as under 5).
  10. Line 117 - It is stated: "over 3,000 GT of the gross registered tonnage". There is a difference between GT and GRT, although GT is used more commonly lately. I suggest writing it as "over 3,000 gross tonnage (GT)" as in VSR.
  11. Figure 2 quality needs to be improved.
  12. Table 2 - It is written "Berth" (singular) instead of "berths" (plural).
  13. Lines 166-167 - There is no need to write "of the gross registered tonnage" since "GT" is already written.
  14. Line 187 - It is written "drought" instead of "draught".
  15. Figure 3 quality needs to be improved.
  16. Line 206 - It is written "Table 5"; shouldn't it be "Table 4"?
  17. In Table 4, in the last row, column Engine age, it is written: "2001-207"; please rectify.
  18. Line 229 - It is written "ragne" instead of "range"; please rectify.
  19. Line 231 - It is written "ηD, ηD and m" instead of "ηD, ηT and m"
  20. Lines 240-242 - The sentence is unclear, please revise it.
  21. Figure 5 quality needs to be improved.
  22. Line 275 - It is stated "Figure 8"; however, it should be "Figure 7".
  23. Figure 8 quality needs to be improved.
  24. Are there any limitations of the study?

I hope that my comments and suggestions will help to improve your paper.

Author Response

Thank you for your valuable comments to improve the paper.

According to the reviewer's comments, the authors reviewed the paper. 

Round 2

Reviewer 1 Report

Comments to the Editor in Chief

Journal: Journal of Marine Science and Engineering.
Manuscript ID: jmse-1617785-V2

Title: Estimation of the Efficiency of Vessel Speed Reduction to Mitigate Gas Emission in Busan Port using AIS Database

Authors: Donghan Woo and Namkyun Im 2*

This manuscript proposed the Spatial analysis of the efficiency of VSR programs is essential to monitor and improve the present program. In the present study, the efficiency of VSR from the Busan Port (BP) including North Port (NP) and Gamcheon Port (GP) was analyzed. A bottom-up activity-based approach using Automatic Identification System (AIS) data is introduced herein for the estimation of spatial marine gas emission in real-time. These authors have proposed the spatial Analysis Domain, AIS Data analysis, and Ship Gas Emission Calculation by Bottom-Up Approach to determine the efficiency of Vessel Speed Reduction, which seems very popular and makes sense.

All of the Figures have been fully written and modified for readers and reviewers easy to understand and review.

I think this manuscript is suitable for publishing in this Journal.

Author Response

Dear reviewer, 

Thank you very much for taking time out of your busy schedule to review the paper.

Thank you again for all your valuable comments to improve the paper.

Best regards,

Donghan Woo and Namkyun Im

Reviewer 2 Report

The authors revised the paper according to the comments and suggestions. However, the quality of figures (besides Figure 3, which was improved) was not improved. Text within figures is now readable, but the figures are still blurry. Please improve it.

Author Response

Dear Reviewer

Thank you very much for taking time out of your busy schedule to review the paper.

Thank you again for all your valuable comments to improve the paper.

Best regards,

Donghan Woo and Namkyun Im
